# The Synergistic Lubrication Effects of h-BN and g-C_3_N_4_ Nanoparticles as Oil-Based Additives for Steel/Steel Contact

**DOI:** 10.3390/ma16144979

**Published:** 2023-07-13

**Authors:** Wen Zhong, Jiazhi Dong, Siqiang Chen, Zhe Tong

**Affiliations:** 1The Key Laboratory of Fluid and Power Machinery, Ministry of Education, Xihua University, Chengdu 610039, China; zw1019@126.com (W.Z.); djz643071849@163.com (J.D.); wdtkbsn@126.com (S.C.); 2Luzhou Laojiao Group Co., Ltd., Luzhou 646000, China; 3School of Mechanical Engineering, North University of China, Taiyuan 030051, China

**Keywords:** oil lubrication conditions, high-temperature tribology, g-C_3_N_4_ and h-BN, synergistic effect

## Abstract

The synergistic effect of different types of solid particles in liquid lubricants is of great interest. In this work, g-C_3_N_4_ nanosheets were initially prepared using a calcination method and then as-prepared, and h-BN were used as lubricating additives to the white oil. A comparison between the mixed additives and the single g-C_3_N_4_ or h-BN additives revealed that the base oil with the addition of g-C_3_N_4_ and h-BN showed the best lubricating properties. The results show a 12.3% reduction in friction coefficient, resulting in a 68.6% reduction in wear rate compared to the white oil when filled with 0.5 wt% g-C_3_N_4_ and h-BN (1:1 by weight). Moreover, the addition of g-C_3_N_4_ and h-BN improves the high-temperature lubrication properties of the white oil. However, the friction coefficient and wear rate increase with increasing oil temperature. The large contact area between g-C_3_N_4_ and its sliding counterpart and the strong adhesive force between h-BN and its sliding counterpart improve the film formation efficiency, leading to enhanced tribological properties under oil lubrication conditions.

## 1. Introduction

Oil lubrication has been widely utilized in various industrial sectors such as lubrication, machining, and heat dissipation, etc., because of its superior film-forming property and liquidity [1,2,3,4]. However, the low carrying capacity of the oil film prevents its low-speed application in heavy duty conditions. Adding solid lubricating particles to the oil lubricant can effectively improve the bearing capacity and lubrication conditions due to the formation of solid tribo-film [5,6,7].

Two-dimensional (2D) graphite-like materials have excellent physical and chemical properties and are usually used as solid additives to improve the tribological properties of oil-based or bulk materials [8,9,10]. The interfacial friction of h-BN can be adjusted and controlled by applying a voltage due to its semiconducting nature and low-friction properties, thus possessing potential as a lubricant additive [11,12]. The formation of tribo-film is considered the main mechanism for enhancing the tribological properties when using a 2D additive [13]. The additive will be absorbed on the surface of the counterpart and grow a protective solid tribo-film, preventing the direct contact of the friction interface during the sliding progress, which is significant in reducing wear and friction, especially under the boundary lubrication conditions [14,15,16]. Graphite carbon nitride (g-C_3_N_4_) is a typical layered structure that has been investigated as a prospective material in varied areas, such as catalysis, CO_2_ reduction, and lubrication [17,18,19]. The film formation efficiency and quality at the sliding interface are the main factors affecting the tribological properties of the lubricant. Additionally, the size, shape, and bonding force between the additives and interface have important effects on the formation of the lubricating film [20].

In recent years, numerous studies have shown that the addition of individual additives does not satisfy engineering requirements. The use of hybrids or multiple particles has attracted attention in tribology design [21,22]. He et al. reported that the use of core–shell Cu@rGO hybrids can significantly improve the mechanical and self-lubricating properties. This was attributed to the prominent load-bearing capacity of Cu NPs and excellent lubrication performance of rGO nanosheets [23]. Li et al. found that the hybridization of CNTs and ZnS improves the dispersion of hybrids in epoxy-based composites. Additionally, the “micro-roller” of CNTs and stress distribution improvement of ZnS show a synergetic effect regarding reductions in friction and wear [24]. Nevertheless, the hybrids typically require chemical or physical reactions to achieve the binding of different particles, which is time-consuming. The direct mixing of different particles in the matrix can realize friction-reducing and wear-resistant properties [25]. Ye et al. reported that MoS_2_ and Zr coating exhibits high hardness and good adhesion strength, improving the tribological behavior and extending the endurance life of the coating [26]. Su et al. found that using graphene oxide (GO) and onion-like carbon (OLC) as lubricating additives can significantly reduce the friction coefficients and wear rates of steel discs. This effect was attributed to the rolling action of OLC and excellent lubricating ability of the GO [27].

For the present study, g-C_3_N_4_ nanosheets were synthesized via a facile one-step calcination method and used together with h-BN to cooperatively modify the tribological properties of white oil under ball–disk sliding conditions at different temperatures. In this paper, the microstructure, components of g-C_3_N_4_- and h-BN, and the underlying lubrication mechanism behind the g-C_3_N_4_- and h-BN-induced enhancements with respect to its tribological properties are systematically described and discussed. Hopefully, this work can provide some guidance for extending the synergetic application of multiple additives in the lubrication field.

## 2. Materials and Experiment

### 2.1. Materials

h-BN with an average size of 2 μm was obtained from Aladdin Reagent (Shanghai) Co., Ltd., Shanghai, China. Urea (CSN_2_H_4_) was purchased from Sinopharm Group, China. White oil was purchased from Aladdin Reagent (Shanghai) Co., Ltd., Shanghai, China. All chemical reagents were used as received without further purification.

### 2.2. Preparation of g-C_3_N_4_

g-C_3_N_4_ nanosheets were synthesized by calcinating urea. Following typical synthesis, 10 g CSN_2_H_4_ was placed in quartz crucible and covered. The reaction was carried out at 550 °C for 5 h in a Muffle furnace with a heating rate of 20 °C/min. Finally, the product was obtained after natural cooling.

### 2.3. Characterization

The crystal phase and microtopography of h-BN and g-C_3_N_4_ nanoparticles were characterized to confirm the synthesis effect. The morphologies of h-BN and g-C_3_N_4_ nanoparticles were characterized by using scanning electron microscopy (SEM, Gemini SEM 500 at 10 kV accelerating voltage, Jena, Germany) and transmission electron microscopy (TEM, Hitachi H800 at 200 kV accelerating voltage, Tokyo, Japan). The crystal phase structures of h-BN and g-C_3_N_4_ nanoparticles were obtained by X-ray diffraction measurements (D8 Advance A25 at 40 kV and 40 mA with Cu Ka). The functional group of h-BN and g-C_3_N_4_ were characterized by Fourier-transform infrared spectroscopy (FTIR Nicolet iS50, Waltham, MA, USA).

### 2.4. Tests and Analysis

A certain quality of particles was added into the white oil and ultrasonically treated for 30 min to obtain a uniform oil-based lubricant, the concentration of soliditives for sliding test was 0.5 wt%. The tribological properties of the oil-based lubricant were evaluated by using a UMT-2 ball-on-disk tribometer (CETR-2, Campbell, CA, USA) with a reciprocating sliding length of 6 mm. The schematic diagram of the contact configuration and the sliding mode is shown in Figure 1. A heating module was placed at the bottom of the plate to achieve different ambient temperatures. The GCr15 bearing steel ball with a diameter of 9.6 mm with surface roughness of Ra = 0.22 μm and bearing steel disk with a diameter of 30 mm, roughness of Ra = 0.1 μm were used as sliding counterparts. The friction tests were carried out at a sliding frequency of 2 Hz with an applied load of 2 N under different temperatures (Room temperature, 50 ± 1 °C and 80 ± 1 °C, 100 ± 1 °C); the sliding duration was set to 1800 s. The ball and disk were cleaned with absolute alcohol prior to testing. Each test was repeated three times, and the average value was used as the final result.

After testing, the wear volume and wear scar were measured and evaluated using a laser scanning confocal microscope (Olympus, Tokyo, Japan, OLS4000) and SEM.

## 3. Results and Discussion

### 3.1. Microstructure of h-BN and g-C_3_N_4_ Particles

Figure 2 shows the microstructural properties of the h-BN and g-C_3_N_4_ particles. It is evident the size of h-BN range from 1 to 3 μm, and the g-C_3_N_4_ exhibit a large lamellar structure with obvious aggregation. Meanwhile, the typical laminated structure of h-BN and g-C_3_N_4_ nanosheets can be seen from the TEM images (Figure 2c,d).

Figure 3 shows the FTIR spectrum of g-C_3_N_4_ and h-BN particles for g-C_3_N_4_; the peaks at 3400–3100 cm^−1^ correspond to the stretching vibrations of N-H. The absorption band of 1700–1100 cm^−1^ correspond to the main characteristic band of g-C_3_N_4_, while the 1650–1630 cm^−1^ band is attributed to N-H shear vibrations and the 1570–1515 cm^−1^ and 1310–1200 cm^−1^ bands are attributed to the N-H bending vibration and C-N stretching vibrations. In addition, the peaks at 880–820 cm^−1^ are attributed to C-H and N-H bending vibrations, indicating the generation of g-C_3_N_4_ [28,29]. For the h-BN, two significant infrared absorption peaks can be seen at 1364 cm^−1^ and 768 cm^−1^, which are attributed to the B-N reticular and B-N-B bending modes, respectively.

XRD was used to characterize the chemical composition and crystal structure of g-C_3_N_4_ and h-BN particles. Figure 4 shows the XRD pattern of the g-C_3_N_4_ nanosheets and the h-BN particles. For the g-C_3_N_4_ nanosheets, the two peaks located at 13.2° and 27.6° are attributed to the (100) and (002) planes, which correspond to the in-plane structural packing and inter-layer structural packing, respectively [30]. Additionally, regarding h-BN particles, the typical diffraction peak located at 26.5° corresponds to the (002) crystal plane of h-BN, which is consistent with the (002) reflection of the hexagonal graphitization [31]. Furthermore, the characteristic peaks located at 41.7° and 76.2° are assigned to the (100) and (110) crystallographic planes of h-BN [32].

### 3.2. Tribological Properties

Figure 5 shows the real-time friction coefficients of oil-based lubricants containing different kinds of solid additives at room temperature. The friction coefficient of the pure oil starts at a very low level and then decreases rapidly with a relatively long running time before reaching final stability. In contrast, the addition of solid additives leads to different levels of reduction in both the friction and running-in time of oil-based lubricant. In particular, the oil-based lubricant shows the lowest friction coefficient with the addition of both h-BN and g-C_3_N_4_, simultaneously. Oil film can form quickly for pure white oil, resulting in a low friction coefficient at the early stages. However, the damage of tribo-film had occurred as the sliding went on due to the low bearing capacity of oil film. With the addition of solid additives, the relative sliding of counterparts was impeded due to an obstacle effect before the additives sufficiently dispersed [33], thus causing an increase in frictional force. However, as the sliding progressed, the solid lubricating film was formed on the surface of the counterparts, therefore reducing the friction coefficient. In addition, the simultaneous presence of h-BN and g-C_3_N_4_ promotes the formation of the solid lubricating film.

Figure 6 shows the average friction coefficient and the corresponding wear rate for different types of oil-based lubricants. It can be established that both the average friction coefficient and wear rate of the three lubricants decrease with the addition of the additive in the white oil, evidencing the excellent lubricating effect of the solid additive. In contrast to h-BN, the counterparts have a lower friction coefficient and wear rate. In addition, the synergistic lubrication effect of h-BN and g-C_3_N_4_ are found to be where the average friction coefficient and wear rate are maximally reduced by 12.3% and 68.6%, respectively, in the presence of both h-BN and g-C_3_N_4_.

Figure 7 shows the SEM images of wear tracks of the disk lubricated with four types of oil-based lubricants. The wear surface was severely damaged with the addition of base oil, as were many deep groves and pits on the disk surface (Figure 7a). The oil film is prone to breaking at a high contact pressure, and direct contact between the counterpart ball and disk occurred, resulting in furrow wear. The width and depth of the wear track and pits decrease with the addition of h-BN to the base oil (Figure 7b). In addition, the g-C_3_N_4_ imbues the base oil with a wear-resistant effect. However, many shallow grooves on the disk surface can still be observed (Figure 7c). The wear surface with g-C_3_N_4_ and h-BN is significantly improved and shows reduced wear width and depth. Additionally, the surface is smoother than those under other lubricants (Figure 7d). In addition, it can be seen from Figure 7d that the main diffraction peak of the XRD pattern indicates the presence of g-C_3_N_4_ and h-BN in the tribo-film, which shows a synergistic effect on friction reduction and improves the wear resistance of g-C_3_N_4_ and h-BN.

Figure 8 shows the 3D morphology of oil lubricants by adding different types of solid additives. The surface of the disk has characteristics that indicate rough wear and shows the deep grooves (Figure 8a). When the h-BN particles were added to the base oil, the width of the wear track decreases significantly. The particles will adhere to the friction surface to prevent direct contact with the sliding counterparts (Figure 8b). As the sliding of the interface continues, the solid lubricating film is formed, thus improving furrow. Compared to that lubricated by h-BN, the addition of g-C_3_N_4_ significantly decreases the width and depth of the wear track, showing an improved anti-wear effect compared to h-BN nanosheets (Figure 8c). The worn surface lubricated by h-BN and g-C_3_N_4_ exhibits a smooth character with only slight grooves being observable; these results are in good agreement with those shown in Figure 7.

### 3.3. Effect of Temperature on the Tribological Properties of Oil Lubricants

Frictional heat accompanies the sliding process, and the temperature has a significant effect on the tribological properties of the liquid lubrication system [34,35]. The above results show that lubricants supplemented with h-BN and g-C_3_N_4_ exhibit the best friction reduction and wear resistance. Therefore, the effect of temperature on the tribological properties of lubricants containing h-BN and g-C_3_N_4_ was further investigated, and the results are shown in Figure 9. The friction coefficients of the two lubricating systems experience a slight increase with elevated temperature, and the lubricant containing h-BN and g-C_3_N_4_ additives still shows a lower friction coefficient than pure white oil at identical temperatures (Figure 9a,b), suggesting that a boundary oil film might be formed on the sliding surface. The bearing capacity of the oil film decreases with an increase in temperature, causing an increase in the friction coefficient [36,37]. Additionally, the wear rate shows the same trend with the friction coefficient (Figure 9c). This seems to indicate that raising the temperature may play a passive role in improving the tribological performances of white oil with additives for ball-disk contact.

SEM images of the worn surface are shown in Figure 10. Clearly, many deep grooves can be found in Figure 10a, and the dominant friction and wear mechanism is typical furrow wear. From Figure 10c,d, it can be seen that the width and depth of the wear tracks increase with increasing temperature. Moreover, plastic deformation can be observed on the surface at 80 °C and 100 °C. In addition to changes in the lubrication conditions, the hardness of the counterparts is an important influencing factor on wear behavior. The hardness of the counterparts decreases with increasing temperature, therefore resulting in severe wear, which may be caused by adhesive wear.

Figure 11 shows the 3D morphologies of the oil lubricants supplemented with h-BN and g-C_3_N_4_ additives at different temperatures. The worn surfaces show increasingly apparent wear tracks with increasing temperature. The wear track at 100 °C shows a wide and shallow profile compared with that at 80 °C. This can be attributed to variations in the contact area of the sliding interface. The hardness decreases as the temperature increases from 80 °C to 100 °C, thus leading to an increase in contact area and resulting in low contact pressure.

### 3.4. Tribological Mechanism

Based on the above-described results and analysis, the reinforcing mechanisms of g-C_3_N_4_ and h-BN additives on the tribological properties of oil-based lubricants is shown in Figure 12. When h-BN particles are used as additives in an oil-based lubricant, the particles tend to move with the flow of oil, resulting in less efficient solid film formation, and the lubricant with h-BN exhibits a relatively high friction coefficient and wear loss. The film formation efficiency may be enhanced when g-C_3_N_4_ nanosheets are added to the lubricant due to their large area. At the same time, the poor adhesion of g-C_3_N_4_ to metal may restrict further improvements in film formation efficiency. When g-C_3_N_4_ and h-BN are added to an oil-based lubricant at the same time, the film formation efficiency and properties can be effectively enhanced compared to single-type addition.

## 4. Conclusions

In summary, g-C_3_N_4_ nanosheets were successfully prepared and subsequently introduced into white oil with h-BN particles to enhance the efficiency of solid lubrication film formation at the sliding interface. The tribological properties of oil-based lubricants after the addition of various additives were investigated at various temperatures, and the corresponding tribological mechanisms were discussed. The main conclusions we have drawn from this study are as follows:(a)g-C_3_N_4_ nanoparticles were successfully prepared, and the nanosheets possess a high specific surface area; the diameter of the g-C_3_N_4_ nanoparticles was microscale.(b)After the addition of the solid additives, both the friction coefficient and wear loss of the disk decrease with the addition of various solid additives compared to the base oil. Additionally, the lubricant supplemented with g-C_3_N_4_ exhibits better tribological properties than that supplemented with h-BN. Furthermore, the oil-based lubricant supplemented with both g-C_3_N_4_ and h-BN (1:1 by weight) showed the lowest friction coefficient of 0.13, and the corresponding disk achieved the lowest wear rate of 0.06 × 10^−5^ mm^3^/(N·m). This can be attributed to the increased formation efficiency of the solid tribo-film due to the synergistic effect of g-C_3_N_4_ and h-BN.(c)However, increasing the temperature resulted in an increase in both wear and the friction coefficient for both the base oil the oil lubricant supplemented with g-C_3_N_4_ and h-BN. The loss of mechanical properties suffered by the sliding counterpart and reduction in oil film bearing capacity with increasing temperature are believed to be the major contributors for this.

## Figures and Tables

**Figure 1 materials-16-04979-f001:**
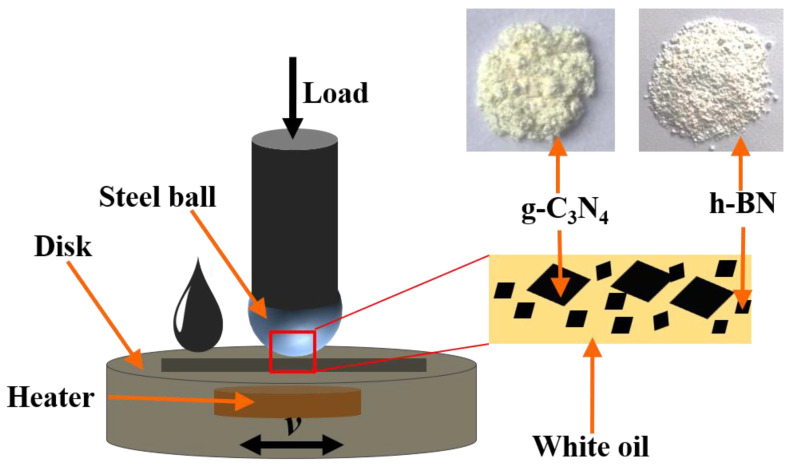
Schematic diagram of contact configuration.

**Figure 2 materials-16-04979-f002:**
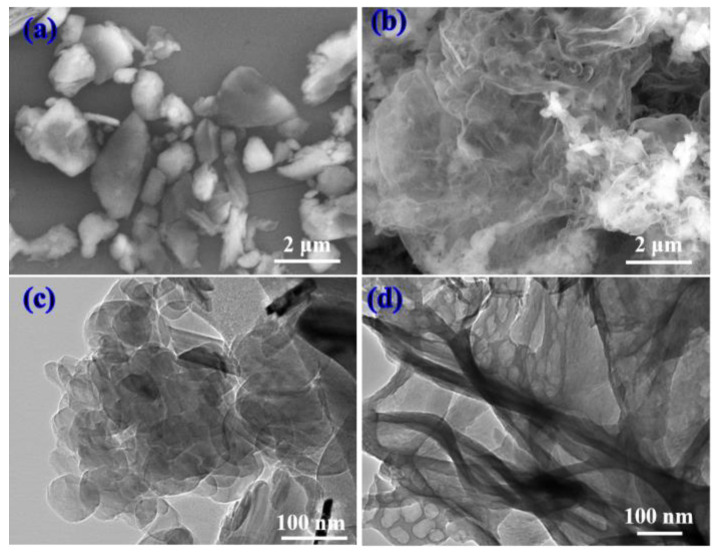
SEM micrographs of (**a**) h-BN nanoparticles and (**b**) g-C_3_N_4_ nanosheets; TEM images of (**c**) h-BN nanoparticles and (**d**) g-C_3_N_4_ nanosheets.

**Figure 3 materials-16-04979-f003:**
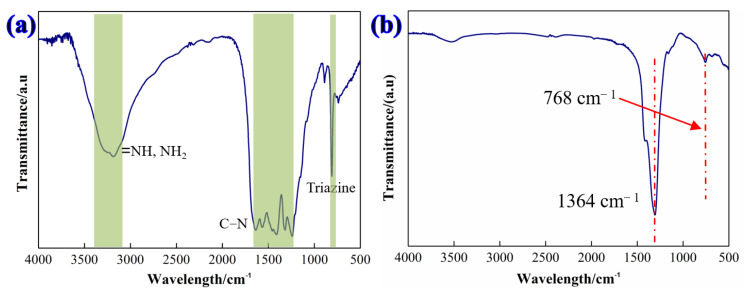
FTIR spectrum of (**a**) as-synthesized g-C_3_N_4_ nanosheets and (**b**) h-BN nanoparticles.

**Figure 4 materials-16-04979-f004:**
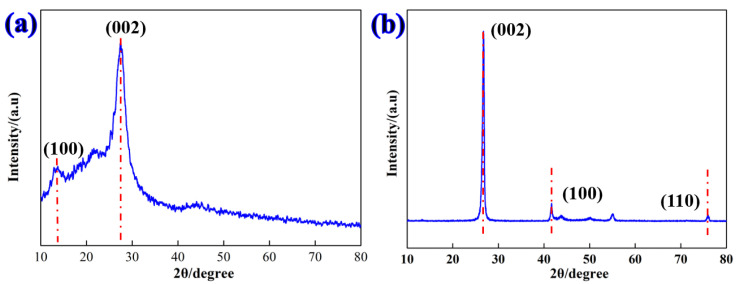
XRD pattern of (**a**) as-synthesized g-C_3_N_4_ nanosheets and (**b**) h-BN nanoparticles.

**Figure 5 materials-16-04979-f005:**
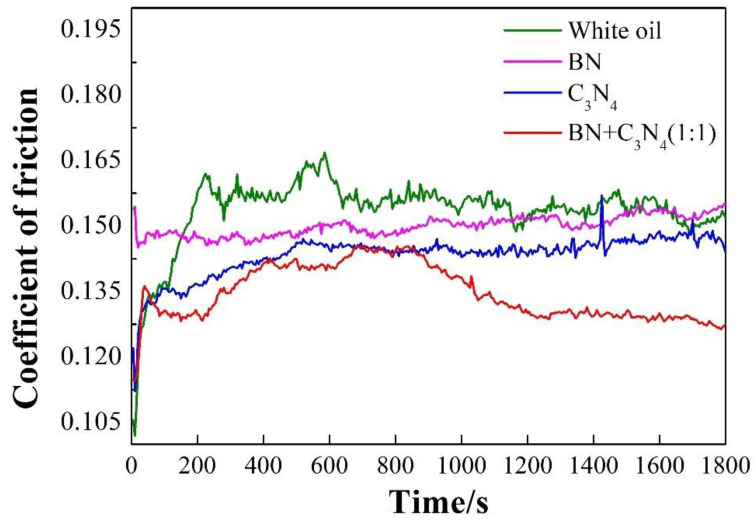
Effect of nano additives on the real-time friction curves of the oil-based lubricant.

**Figure 6 materials-16-04979-f006:**
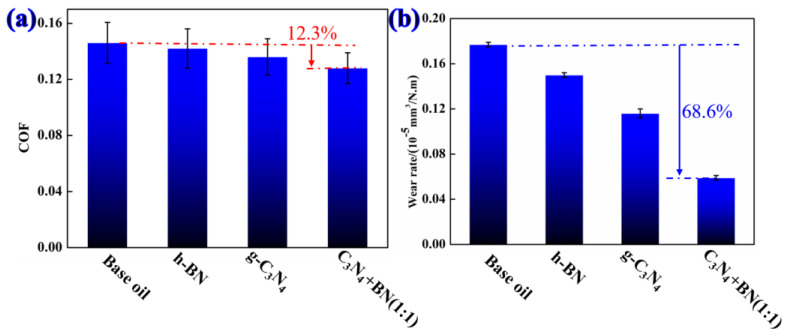
(**a**) Average friction coefficient and (**b**) wear rate of white oil containing different types of additives.

**Figure 7 materials-16-04979-f007:**
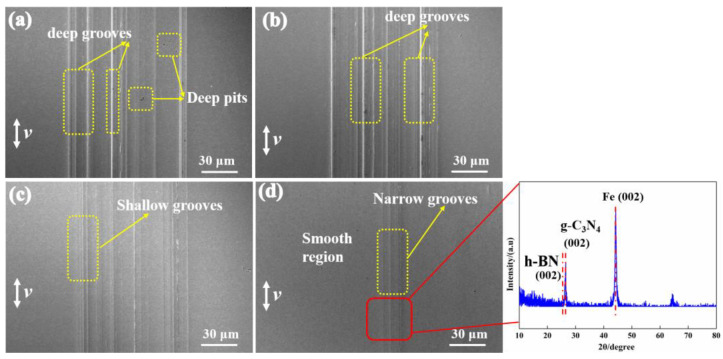
SEM images of the worn surfaces of the disk lubricated by (**a**) pure white oil, (**b**) white oil with h-BN, (**c**) white oil with g-C_3_N_4_, and (**d**) white oil with h-BN and g-C_3_N_4_, as well as the corresponding XRD pattern.

**Figure 8 materials-16-04979-f008:**
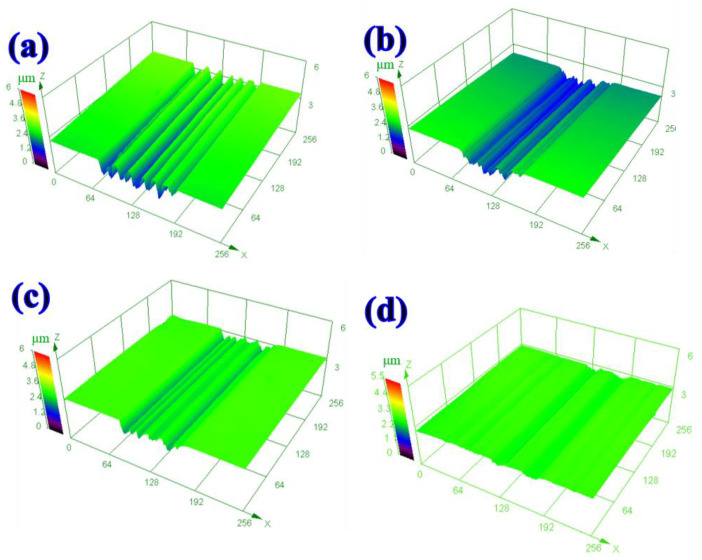
Three-dimensional morphologies of worn tracks lubricated by: (**a**) Pure white oil, (**b**) white oil with h-BN, (**c**) white oil with g-C_3_N_4_, and (**d**) white oil with h-BN and g-C_3_N_4_.

**Figure 9 materials-16-04979-f009:**
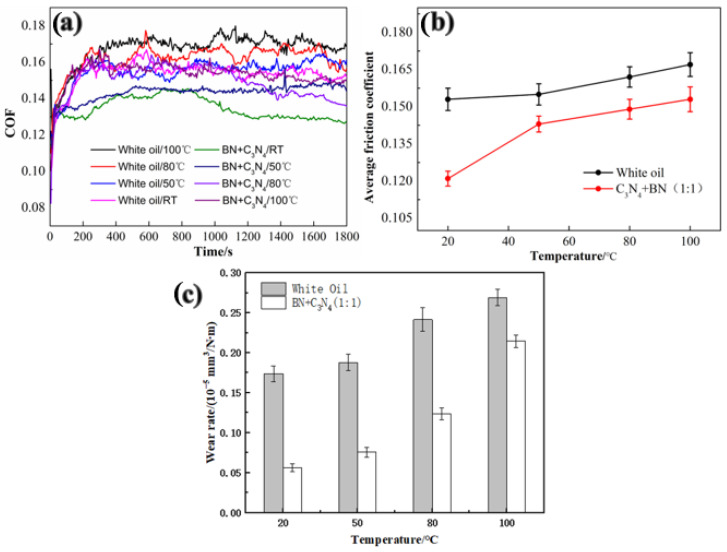
(**a**) Real-time friction curve, (**b**) average coefficient of friction, and (**c**) wear rate of pure white oil and white oil with adding of h-BN and g-C_3_N_4_ at different temperatures.

**Figure 10 materials-16-04979-f010:**
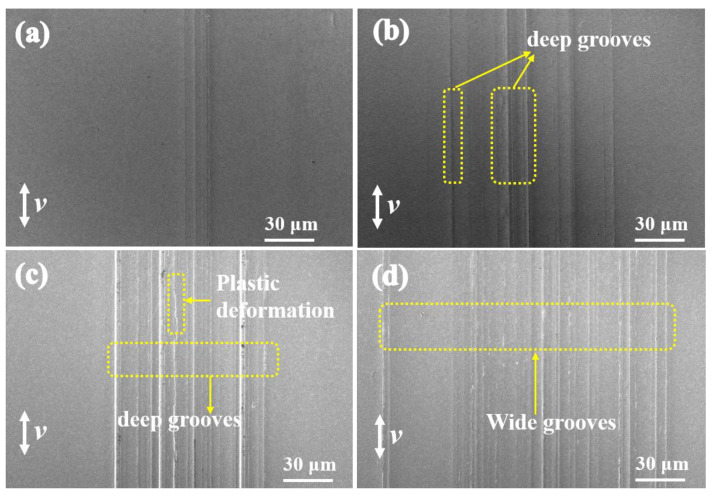
SEM images of the worn surfaces of the disk lubricated by white oil with h-BN and g-C_3_N_4_ additives at (**a**) room temperature, (**b**) 50 °C, (**c**) 80 °C, and (**d**) 100 °C.

**Figure 11 materials-16-04979-f011:**
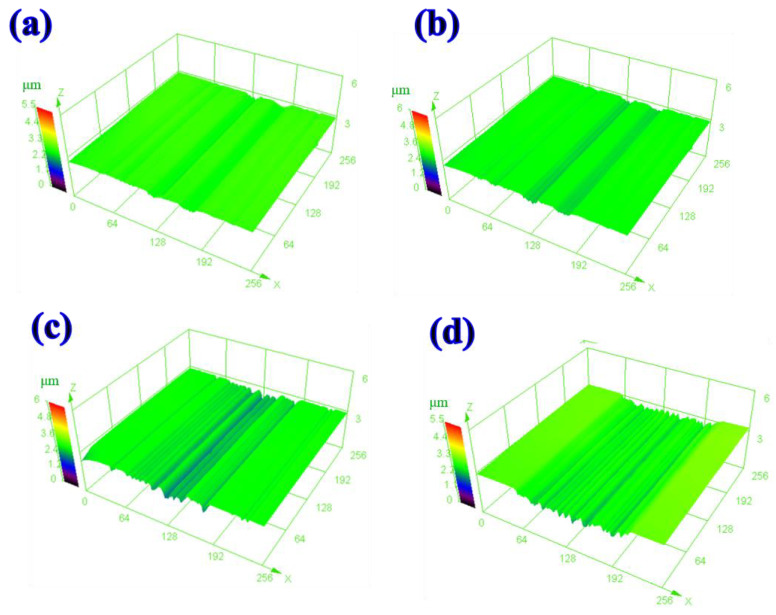
Three-dimensional morphologies of worn tracks lubricated by white oil with h-BN and g-C_3_N_4_ additives at (**a**) room temperature, (**b**) 50 °C, (**c**) 80 °C, and (**d**) 100 °C.

**Figure 12 materials-16-04979-f012:**
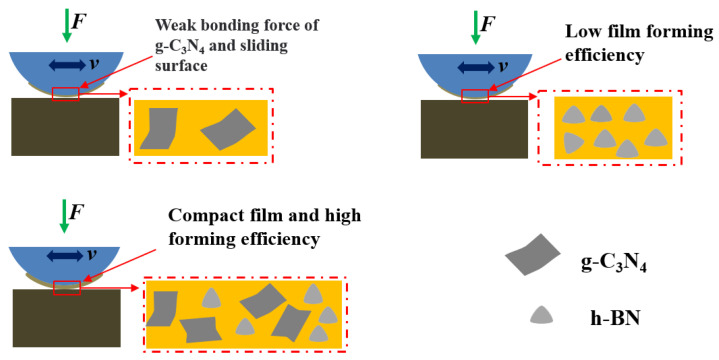
Reinforcing mechanism of g-C_3_N_4_ and h-BN additives on the tribological properties of oil-based lubricants.

## Data Availability

The data presented in this study are available on request from the corresponding author.

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
