# Peer review of "The Synergistic Lubrication Effects of h-BN and g-C_3_N_4_ Nanoparticles as Oil-Based Additives for Steel/Steel Contact"

_materials, 2023, doi:10.3390/ma16144979_

Round 1

Reviewer 1 Report

1. which factors improve the film forming efficiency and  leads to enhanced tribological properties?

2. Why friction coefficient and wear loss of disk are decreased with the addition of various solid additives ?

3. How increasing the temperature results in increased wear

4. Literature review is to be improved. Add the following literature in the revised manuscript. https://doi.org/10.1177/0954406212466193

5. English correction required

Average

Author Response

Question 1): Which factors improve the film forming efficiency and leads to enhanced tribological properties?

Answer 1): Thank you very much for your valuable comment.

The particles with large size or aggregations accumulated in front of the contact area tended to enter the contact area and form a well-distributed tribo-film on the contact surface, which were beneficial in reducing COF and wear loss[1].

Herein, the large superficial area of g-C3N4 nanosheets and strong bonding force of h-BN with metal sliding counterpart contribute to continuous tribo-film forming. The solid tribo-film with stratified structure was useful for improvement of tribological properties.

Reference

[1]    Wu H, Qin L, Dong G, et al. An investigation on the lubrication mechanism of MoS2 nano sheet in point contact: The manner of particle entering the contact area[J]. Tribology International, 2017,107:48-55.

Question 2): Why friction coefficient and wear loss of disk are decreased with the addition of various solid additives?

Answer 2): Thanks so much for your valuable comment.

The oil film on the surface of counterparts tend to break during the sliding progress due to the low bearing capacity. However, with the addition of solid additives, the solid tribo-film with lamellar structure could form on the surface of sliding interface, the friction coefficient and thus the wear rate is reduced because of the high bearing capacity and low Interlaminar shear force.

Question 3): How increasing the temperature results in increased wear

Answer 3): Thank you very much for your valuable comment.

       The reason can be attributed to the variation in the mechanical properties of the substrate and the bearing capacity of the film. The hardness of sliding counterpart decreases as the temperature increased, on the other hand, the bearing capacity of oil film decreases with increasing temperature. As a result, the wear rate increases.

Question 4): Literature review is to be improved. Add the following literature in the revised manuscript. https://doi.org/10.1177/0954406212466193

Answer 4): Thank you very much for your valuable comment.

In the literature review stage, we ignored this valuable information. Now we are really inspired by these articles. We have learned a lot about the application fields, synthesis methods, and construction of nano-particles, which contribute to us developing nano-particles reinforced oil-based lubrication. The article has been added in Section 1 in revised manuscript and marked in blue. The corresponding reference numbers are 4.

Question 5): English correction required.

Answer 5): Thank you very much for your valuable comment.

We feel sorry for our poor writings. To address this issue, we do invite a friend of us who is a native English speaker help polish our article. Due to our friend’s help, the article has a better readability. The modifications of language and format have been marked in blue in revised manuscript.

Reviewer 2 Report

Besides indicated corrections (directly given in comment boxes) you will find comments or questions (in angle brackets) in the commented PDF file (attached). The present text would have to be adapted or supplemented accordingly. Appropriate explanations would have to be inserted for the sake of clarity and comprehensibility.
Three weaknesses should be pointed out here:

1. The introduction is to be completed with details! Since friction and wear are system properties it is not sufficient for a professional classification of the state of the art to cite some publications only indicating "friction-reducing and wear resistance properties", "reduce the friction coefficients and wear rates of steel disc", and similar, when the system conditions (structure and stresses) are not clear (to the readers)!

2. It is questionable whether the plastic deformation simply caused by the temperature increase described. It would be helpful to consider the relative normal force ((Hertzian) pressure as a consequence of 2 N normal force and the strength properties (stress-strain characteristic, also as a function of temperature) of the sample materials used.

3. The interpretation of studied tribologic processes reads plausibly. However, is it evidenced by surface analyses of the samples after the tests, e. g. by making a comparison with the spectra as presented in chapter 3.1? Such an addition seems unavoidable, especially since the observed effects are presented as "film formation".

Moderate editing of English language (wording, grammar) is required.

Author Response

Besides indicated corrections (directly given in comment boxes) you will find comments or questions (in angle brackets) in the commented PDF file (attached). The present text would have to be adapted or supplemented accordingly. Appropriate explanations would have to be inserted for the sake of clarity and comprehensibility. Three weaknesses should be pointed out here.

Question 1). The introduction is to be completed with details! Since friction and wear are system properties it is not sufficient for a professional classification of the state of the art to cite some publications only indicating "friction-reducing and wear resistance properties", "reduce the friction coefficients and wear rates of steel disc", and similar, when the system conditions (structure and stresses) are not clear (to the readers)!

Answer 1): Thank you very much for your valuable comment.

The review of system properties of friction and wear does help to understand the state of the art relevant to solid additives in oil-based lubricant. The content about system properties of friction and wear as well as working conditions have been modified and added in section of introduction and marked in blue in revised manuscript.

Question 2). It is questionable whether the plastic deformation simply caused by the temperature increase described. It would be helpful to consider the relative normal force ((Hertzian) pressure as a consequence of 2 N normal force and the strength properties (stress-strain characteristic, also as a function of temperature) of the sample materials used.

Answer 2): Thank you very much for your valuable comment.

       In practice, the plastic deformation tends to occur in the initial phase of contact at different temperatures due to the high contact pressure. (The load of 2 N corresponds to maximum Hertz contact pressure of about 590 Mpa. The results based on mechanical properties of counterpart: Bearing steel with elastic modulus of 210 Gpa and Poisson ratio of 0.29 in this study). Then, the plastic deformation will decrease or disappear because of the large contact size. The temperature does have an important effect on the stress-strain characteristic of sliding counterpart, and the plastic deformation increased with the increases of temperature.

Question 3) The interpretation of studied tribological processes reads plausibly. However, is it evidenced by surface analyses of the samples after the tests, e. g. by making a comparison with the spectra as presented in chapter 3.1? Such an addition seems unavoidable, especially since the observed effects are presented as "film formation".

Answer 3): Thank you very much for your valuable comment

           The analysis of tribo-film on the surface of sliding interface can help to reveal the friction mechanism. The component and morphology analysis of tribo-film have been conducted in section 3.2 in revised manuscript and marked in blue.

Reviewer 3 Report

The authors have written a manuscript entitled, "The synergistic lubrication effects of h-BN and g-C3N4 nano-particles as oil-based additives for steel/steel contact". The manuscript has shown a good research result. The manuscript can be considered for acceptance after the following few points:

1. Could the authors elaborate more on the results by comparing any previous results with similar research, if any?

2. Did the authors confirm the element types in the samples using EDS/EDX or any similar characterization besides using XRD?

Author Response

The authors have written a manuscript entitled, "The synergistic lubrication effects of h-BN and g-C3N4 nano-particles as oil-based additives for steel/steel contact". The manuscript has shown a good research result. The manuscript can be considered for acceptance after the following few points:

Question 1): Could the authors elaborate more on the results by comparing any previous results with similar research, if any?

Answer 1): Thank you very much for your valuable comment.

It is instructive to compare research results with previous results, which contribute to the illustrating of frictional characteristics and mechanism. The detailed discussions of this part have been conducted in Section 3 in revised manuscript and marked in blue.

Question 2): Did the authors confirm the element types in the samples using EDS/EDX or any similar characterization besides using XRD?

Answer 2): Thank you very much for your valuable comment.

EDS is an effective method to characterize the component of the film. Herein, the samples were characterized by using XRD and FT-IR, and the results are presented to verify the structure and component of h-BN and g-C3N4. Therefore, we generally consider the solid additives that were prepared and identified successfully.

Round 2

Reviewer 2 Report

Please consider the comments/remarks directly given in the commented file of the v1-version.

Note: Due to some changes in the v2-version some of my comments/remarks are already obsolete!

Please consider the respective comments/remarks directly given in the commented file of the v1-version.

Author Response

Besides indicated corrections (directly given in comment boxes) you will find comments or questions (in angle brackets) in the commented PDF file (attached). The present text would have to be adapted or supplemented accordingly. Appropriate explanations would have to be inserted for the sake of clarity and comprehensibility. Three weaknesses should be pointed out here.

Question 1). The introduction is to be completed with details! Since friction and wear are system properties it is not sufficient for a professional classification of the state of the art to cite some publications only indicating "friction-reducing and wear resistance properties", "reduce the friction coefficients and wear rates of steel disc", and similar, when the system conditions (structure and stresses) are not clear (to the readers)!

Answer 1): Thank you very much for your valuable comment.

The review of system properties of friction and wear does help to understand the state of the art relevant to solid additives in oil-based lubricant. The content about system properties of friction and wear as well as working conditions have been modified and added in section of introduction and marked in blue in revised manuscript.

Question 2). It is questionable whether the plastic deformation simply caused by the temperature increase described. It would be helpful to consider the relative normal force ((Hertzian) pressure as a consequence of 2 N normal force and the strength properties (stress-strain characteristic, also as a function of temperature) of the sample materials used.

Answer 2): Thank you very much for your valuable comment.

       In practice, the plastic deformation tends to occur in the initial phase of contact at different temperatures due to the high contact pressure. (The load of 2 N corresponds to maximum Hertz contact pressure of about 590 Mpa. The results based on mechanical properties of counterpart: Bearing steel with elastic modulus of 210 Gpa and Poisson ratio of 0.29 in this study). Then, the plastic deformation will decrease or disappear because of the large contact size. The temperature does have an important effect on the stress-strain characteristic of sliding counterpart, and the plastic deformation increased with the increases of temperature.

Question 3) The interpretation of studied tribological processes reads plausibly. However, is it evidenced by surface analyses of the samples after the tests, e. g. by making a comparison with the spectra as presented in chapter 3.1? Such an addition seems unavoidable, especially since the observed effects are presented as "film formation".

Answer 3): Thank you very much for your valuable comment

           The analysis of tribo-film on the surface of sliding interface can help to reveal the friction mechanism. The component and morphology analysis of tribo-film have been conducted in section 3.2 in revised manuscript and marked in blue.

Question 4) About formatting, writing and chart issues.

Answer 4): Thank you very much for your valuable comment

We are very sorry for so many clerical errors. The corresponding modifications have been conducted in revised manuscript of the v1-version.

Question 5)

  • What exyctly is meant with "average size"? Are there (other) shape parameters to be mentioned (Line 70)

  Answer): As shown in Fig.2a, the size of h-BN is non-uniform, therefore, it is expressed using "average".

  • "repeated three times" means altogether four tests each? (Line 100)

Answer): Yes, altogether test was repeated three times.

  • Is the sentence correct? What is meant with "running time"? (Line 143)

Answer): It should be "running-in" time

  • It is supposed that the curves do not represent average values! (Line 155)

Answer): Yes, the Fig.5 present the real-time results, not average values. The average values are shown in Fig.6.
